# Facial Signals and Social Actions in Multimodal Face-to-Face Interaction

**DOI:** 10.3390/brainsci11081017

**Published:** 2021-07-30

**Authors:** Naomi Nota, James P. Trujillo, Judith Holler

**Affiliations:** 1Donders Institute for Brain, Cognition, and Behaviour, 6525 AJ Nijmegen, The Netherlands; j.trujillo@donders.ru.nl (J.P.T.); j.holler@donders.ru.nl (J.H.); 2Max Planck Institute for Psycholinguistics, 6525 XD Nijmegen, The Netherlands

**Keywords:** facial signals, social actions, questions, responses, intentions, multimodal communication, conversation, turn-taking

## Abstract

In a conversation, recognising the speaker’s social action (e.g., a request) early may help the potential following speakers understand the intended message quickly, and plan a timely response. Human language is multimodal, and several studies have demonstrated the contribution of the body to communication. However, comparatively few studies have investigated (non-emotional) conversational facial signals and very little is known about how they contribute to the communication of social actions. Therefore, we investigated how facial signals map onto the expressions of two fundamental social actions in conversations: asking questions and providing responses. We studied the distribution and timing of 12 facial signals across 6778 questions and 4553 responses, annotated holistically in a corpus of 34 dyadic face-to-face Dutch conversations. Moreover, we analysed facial signal clustering to find out whether there are specific combinations of facial signals within questions or responses. Results showed a high proportion of facial signals, with a qualitatively different distribution in questions versus responses. Additionally, clusters of facial signals were identified. Most facial signals occurred early in the utterance, and had earlier onsets in questions. Thus, facial signals may critically contribute to the communication of social actions in conversation by providing social action-specific visual information.

## 1. Introduction

A crucial prerequisite for having a successful conversation is to recognise the speaker’s social action, or what an utterance does in a conversation. For instance, this could be a request, an offer, or a suggestion [1] (in some ways ‘social actions’ are similar to the notion of speech acts [2,3]). Conversation is a time-pressured environment, consisting of minimal gaps and overlaps between interlocutors [4,5,6,7]. This is especially true for responses to questions, since a long gap is meaningful by itself, and may indicate a dispreferred response [8]. Thus, recognising the speaker’s social action early may effectively constrain the possibilities of what the speaker is going to say; thus, helping potential following speakers to more quickly understand the intended message and plan a timely response in return [9,10,11,12].

Human language is a multimodal phenomenon (e.g., [13,14,15,16,17,18,19]), and by now, a substantial number of studies have demonstrated the contribution of the body to communication [20]. However, comparatively few studies have investigated (non-emotional) visual signals coming from the speaker’s face, and very little is known about how facial signals contribute to the communication of social actions.

### 1.1. The Role of Facial Signals in (Non-Emotional) Communication

Although facial signals have been studied most in the context of emotion expression, some studies have investigated facial signalling in connection with semantic and pragmatic functions in talk. For example, facial signals can act as grammatical markers (e.g., emphasisers [21,22,23]) or mark the organizational structure of the speech (e.g., begin, end, or continuation of topic [22,24]). Several studies have also found associations between facial signals and social actions. For instance, mouth movements, such as smiles, can indicate an ironic or sarcastic intent, in combination with direct gaze and head movements, among other signals [25,26,27]. This is comparable to sign language, where mouth corners up or down can signal the signer’s ironic meaning and attitude when they do not match with the utterance meaning [28]. Further, in spoken language, smiles have been shown to foreshadow an emotional stance [29], and eyebrow frowns to announce a problematic aspect in the topic of conversation [30]. Eyebrow frowns have also been associated with addressees signalling a need for clarification [23,31,32,33]. Conversely, addresses also use facial signals to signal understanding, such as with long blinks [32,34,35]. Other facial signals that were observed to act as backchannels are eyebrow raises, and mouth movements, such as pressed lips, mouth corners down [22], and smiles [36]. These visual backchannels may help the addressee provide feedback to the speaker.

Similarly, combinations of co-occurring facial signals have been associated with specific social actions. The facial expression (i.e., meaningful assembly of facial signals) referred to as the *not-face* has been linked to negative messages during conversation and was argued to communicate negation or disagreement [37]. The not-face consists of a combination of signals associated with the expression of anger, disgust, and contempt, and typically includes eyebrow frowns, compressed chin muscles, and pressed lips (as well as squints and nose wrinkles; however, these were not found to be consistently active). This expression of negation has been observed across different languages, with or without speech, and in sign language [37]. By using this expression, a speaker may help the next speaker recognise what the social action of the utterance will be. For example, signals belonging to the not-face may help to indicate that the speaker will take the floor with a message that is not in alignment with prior speech, thus making the speaker’s social action more transparent to the receiver (who, in turn, may be able to prepare a fitting response early). The same holds for the *facial shrug*, which consists of an eyebrow raise and pulled down mouth corners. This facial expression signals indifference or lack of knowledge in a similar way to shoulder shrugs [21,25,38], and may indicate that the speaker is disinterested in a certain conversational topic. Furthermore, the *thinking-face*, consisting of a short gaze shift (away from the speaker) or closure of the eyes, can act as a signal that expresses effort while thinking of what to say, remembering something, or searching for a word or concept [25,39]. The thinking-face has been found to often occur during periods of silence at the beginning of a topic [22]. The expression may indicate that the speaker wants to keep the floor until they have remembered what they were searching for, or may announce that they do not know something. Thus, there is clear evidence that specific facial signals, or combinations of facial signals, can contribute to signalling specific social actions in conversation. However, it is currently unclear whether specific facial signals, as well as combinations of facial signals, map onto the fundamental conversational social actions of questioning and responding.

### 1.2. Facial Signals as Markers of Questions and Responses

Questions and responses are an important focus, for one, because they are foundational building blocks of conversation [7,40]. For another, the normative principles by which interlocutors abide (at least in Western interactions) make responses to questions rather mandatory, and they need to be swift (unless they are dispreferred [8]). Fast social action recognition is therefore particularly relevant for question turns.

Several facial signals have been linked to questions and responses in previous research. The eyes have been found to play a role in signalling questions and responses, albeit in different ways. Direct gaze has been linked to questions in both spoken and signed languages, where it is often held until a response is provided by the addressee [41,42,43,44,45,46] and has been argued to fulfil response mobilising functions [45]. The next speaker may in turn signal dispreferred responses performing gaze shifts away from the addressee [47]; therefore, gaze shifts may be generally quite common signals in responses.

Like for gaze, many studies have found links between eyebrow movements, such as frowns, raises, and questions in spoken and signed languages [21,22,23,32,33,42,46,48,49,50,51,52,53,54,55,56]. However, a few studies did not find evidence for eyebrow movements distinguishing questions from other types of social actions [24,57]. On the contrary, eyebrow raises were found to be more frequent in instructions compared to questions involving requests or in acknowledgments of information [24]. Thus, although it seems that eyebrow movements do play a role in the signalling of questionhood based on a number of studies, extant evidence is partly discrepant, making larger scale, systematic investigations necessary. Such investigations are also needed to investigate the co-occurrence of eyebrow movements with other facial signals in the context of marking questionhood, which we currently know very little about. Moreover, the existing literature is partly based on scripted behaviour, underlining the need for systematic analyses of eyebrow movements in naturalistic conversational interactions.

### 1.3. Facial Signal Timing and Early Processing

Another critical component in addition to the form of the facial signals (contributing to social action recognition in terms of the ‘what’) is the timing of such signals (i.e., contributing to the ‘when’) for fast social action recognition in a turn-taking context. Questions with manual gestures and/or head gestures have been found to result in faster responses than questions without such gestures [58,59], suggesting multimodal facilitation. Even in the absence of speech, a manual action can offer a direct perceptual signal for the observer to read the producer’s communicative goal (e.g., [60,61]). This multimodal signalling facilitation may also occur for facial signals, since they may reflect what social action the utterance is performing. If facial signals indeed serve as facilitators of social action recognition, then they should occur relatively early in the turn where they may exert the greatest influence on early social action attribution. Early social action recognition could potentially allow the next speaker to understand the intended message more quickly, thus ensuring that they have sufficient time to plan their utterance [9,10,11,12]. There have been some reports of facial signals occurring early in the utterance and, thus, foreshadowing the social action of the utterance, such as turn-opening smiles, frowns [29,30], and gaze shifts away from the addressee [47]. However, there are also reports of facial signals systematically occurring late in an utterance, for example when speakers convey irony by smiling at the end of their speech, as a way to make it explicit that they are checking if the ironic message is understood [27]. Thus, it could be that many facial signals occur early in order to facilitate early recognition of the social action, but it could be that particular facial signals occur late in specific cases. The timing of facial signals within the verbal utterances with which they occur has received very little attention, leaving it an open question whether these signals are contributing to *early* recognition of the social action.

### 1.4. Current Study

To address the outstanding issues and questions highlighted above, the current study aimed to investigate a wide range of facial signals in multimodal face-to-face interaction, using a rich corpus of dyadic Dutch face-to-face conversations. We asked how the production of different facial signals mapped onto the communication of two fundamental social actions in conversation: asking questions and providing responses. To our knowledge, this is the first systematic investigation of conversational facial signals on such a large dataset and for these two specific social actions. The research questions we addressed were as follows:(1)Which facial signals occur with questions and responses, and what are their distributions across questions and responses?(2)How do facial signals cluster, and are there specific *combinations* of co-occurring facial signals that map onto questions and responses?(3)What are the timings of facial signals within questions and responses?

Due to the exploratory nature of the study and the relative paucity of research on facial signalling of social actions, we were only able to make predictions about a different distribution in questions versus responses for eyebrow movements (a domain with extant findings from at least a few studies), but did not make predictions about other facial signals. In line with studies showing eyebrow frowns and raises functioning as question markers, we hypothesised that they would occur more in questions versus responses [21,22,23,32,33,42,46,48,49,50,51,52,53,54,55,56]. We also expected that facial signals belonging to the same complex facial expressions, such as the not-face [37], facial shrug [21,25,38], and thinking-face [25,39] would co-occur, since these are known patterns in the literature. How often they would occur with questions and responses, however, is an open question we aimed to answer.

In agreement with the idea of early signalling as a facilitator of early action recognition in conversational interaction [9,10,11,12], we hypothesised that most facial signals would occur around the start of the utterance. However, other factors could potentially determine the timing of facial signals as well. It could be that facial signals, such as eyebrow movements, occur most at the start and/or the end of the utterance because they indicate turn boundaries [22], or mark the organizational structure of a topic [22,24]. We expect that such effects will be evident in both questions and responses equally and, thus, any early timing associated with the facilitation of social action formation and recognition should still be evident in the data. Results from this study provide more insights into the specific association between facial signals and social actions in multimodal face-to-face interaction, and how they may contribute to early processing of an utterance. Our study will also be informative to research that seeks to investigate the cognitive and neural basis of social action recognition. There is evidence that visual signals are integrated with speech [62,63]; however, brain responses to social actions have mostly been investigated without including the visual modality [10,11,64], leaving it an open question whether, and if so how, facial signals contribute to social action comprehension.

## 2. Materials and Methods

### 2.1. Corpus

This study used 34 video dyads that form part of a multimodal Dutch face-to-face conversation corpus (CoAct corpus, ERC project led by JH). The videos consisted of Dutch native speaker pairs of acquaintances (*M* age = 23.10, *SD* = 8, 51 female, 17 male), without motoric or language problems, and with normal or corrected-to-normal vision, holding a dyadic casual conversation for one hour while being recorded.

The recording session consisted of three parts, each lasting 20 min, to increase the likelihood of eliciting different social actions. In the first 20 min, participants held a free, entirely unguided conversation. During the second part, participants discussed one out of three themes: privacy, social media, or language in teaching. They were instructed to share their opinions about these themes and to discuss their agreements and disagreements per theme. Before starting with the second part, participants read some examples of the themes (Appendix A). If they finished discussing one theme, they could pick another. During the third part, participants were asked to think of their ideal holiday affordable with their own budget. They were given two minutes to think and write their ideas down on a piece of paper, after which they discussed them with their partner with the aim to come to a joint holiday plan which they would both enjoy.

Prior to each of the three parts, participants held a T-pose for three seconds to calibrate the motion tracking software (Kinect for Windows 2, Brekel Pro Face 2.39, Brekel Pro Body 2.48, and Wireshark) and clap to be able to synchronise audible and visible information (the kinematic data are not analysed in the current study).

Informed consent was obtained before and after filming. Participants were asked to fill in a demographics questionnaire prior to the study, and four questionnaires at the end of the study. These contained questions about the relationship between the conversational partners and their conversation quality, the Empathy Quotient [65], the Fear of Negative Evaluation scale [66], and a question assessing explicit awareness of the experimental aim. Information from these questionnaires was not used in the current study. Participants were rewarded with 18 euros at the end of the session. The corpus study was approved by the Ethics Committee of the Social Sciences department of the Radboud University Nijmegen (ethic approval code is ECSW 2018-124).

### 2.2. Apparatus

The conversations were recorded in a soundproof room at the Max Planck Institute for Psycholinguistics in Nijmegen, The Netherlands. Participants were seated facing each other at approximately 90 cm distance from the front edge of the seats (Figure 1).

Two video cameras (Canon XE405) were used to record frontal views of each participant, two cameras recorded each participant’s body from a 45 degree angle (Canon XF205 Camcorder), two cameras (Canon XF205 Camcorder) recorded each participant from a birds-eye view while mounted on a tripod, and finally one camera (Canon Legria HF G10) recorded the scene view, displaying both participant at the same time. All cameras were recorded at 25 fps. Audio was recorded using two directional microphones (Sennheiser me-64) for each participant (see the Appendix A for an overview of the set-up). Each recording session resulted in seven video files and two audio files, which were synchronised and exported as a single audio-video file for analysis in Adobe Premiere Pro CS6 (MPEG, 25 fps), resulting in a time resolution of approximately 40 ms, the duration of a single frame. For the coding of facial signals reported in the present study, only the face close-ups were used, one at a time for best visibility of detailed facial signals.

### 2.3. Transcriptions

#### 2.3.1. Questions and Responses

The analysis focused on questions and responses. First, an automatic orthographic transcription of the speech signal was made using the Bavarian Archive for Speech Signals Webservices [67]. Questions and responses were identified and coded in ELAN (5.5; [68]), largely following the coding scheme of Stivers and Enfield [69]. In addition to this scheme, more rules were applied on an inductive basis, in order to account for the complexity of the data in the corpus. Specifically, a holistic approach was adopted, taking into consideration visual bodily signals, context, phrasing, intonation, and addressee behaviour. Any verbal response to a question was transcribed, including conventionalised interjections such as “uh” or “hmm”. Any non-verbal sounds were excluded (e.g., laughter, sighs). This was done by two human coders, one native speaker of Dutch, and one highly proficient speaker of Dutch. The interrater reliability between the two coders was calculated with raw agreement [70,71] and a modified Cohen’s kappa using EasyDIAg [72] on 12% of the total data (4 dyads, all parts). EasyDIAg is an open-source tool that has been used as a standard method for calculating a modified Cohen’s kappa. It is based on the amount of temporal overlap between transcriptions, categorization of values, and segmentation of behaviour. A standard overlap criterion of 60% was used, meaning that there should be a temporal overlap of 60% between events. Reliability between the coders resulted in a raw agreement of 75% and *k* = 0.74 for questions, and a raw agreement of 73% and *k* = 0.73 for responses, indicating substantial agreement. The precise beginnings and endings of the question and response transcriptions were segmented using Praat (5.1; [73]) based on the criteria of the Eye-tracking in Multimodal Interaction Corpus (EMIC; [58,74]). This resulted in a total of 6778 questions (duration *Mdn* = 1114, *min* = 99, *max* = 13,145, *IQR =* 1138, in ms) and 4553 responses (duration *Mdn* = 1045, *min =* 91, *max =* 18,615, *IQR =* 1596, in ms).

#### 2.3.2. Facial Signals

For the present analyses, facial signals were annotated in ELAN (5.5; [68]) based on the synchronised frontal view videos from the CoAct corpus and linked to the question and response transcriptions. Only facial signals that started or ended between a time window of 200 ms before the onset of the question and response transcriptions and 200 ms after the offset of the question and response transcriptions were annotated (until their begin or end, which could be outside of the 200 ms time window). The manual annotations were created on a frame-by-frame basis, one tier at a time, by five trained human coders, all native speakers of Dutch.

Facial signals were all annotated except if they involved movements that obviously did not carry some sort of communicative meaning related to the questions or responses, as we were interested in the communicative aspect instead of the pure muscle movements. Like for questions and responses, the context of the conversational exchange was taken into account, to estimate the communicative meaning. Movements due to swallowing, inhaling, laughter, or articulation were not considered. Facial signals coded consisted of: eyebrow movements (frowns, raises, frown raises, unilateral raises, lowering), eye widenings, squints, blinks, gaze shifts (gaze away from the addressee, position of the pupil), nose wrinkles, and non-articulatory mouth movements (pressed lips, corners down, corners back, smiles) (see the Appendix A for example frames per facial signal). The exclusion of other facial signals was based on economic considerations; however, this does not mean that they are not informative in conversation. Facial signals produced by participants in the addressee role were not annotated, as we aimed to investigate the relation between facial signals and social actions produced by speakers of questions and responses only.

The signals were annotated from the first evidence of movement until the respective articulator moved back into neutral position. Visual behaviour can start before or last longer than the actual verbal message, due to the way visual signals and speech are produced in natural conversation; therefore, facial signals were coded from where they started until they ended, except when speech not forming part of the question or response in question began, or when laughter (without speech) occurred. In those cases, the annotation lasted until the first evidence, or begun after the last evidence, of speech not related to the questions/responses or laughter.

To avoid artefacts from potential timing discrepancies in ELAN between audio and image, any facial signal annotation that started or ended within 80 ms (two frames) of an unrelated speech boundary was excluded from the analysis. This prevented including facial signals that were related to any other speech from the speaker instead of the question or a response. Any facial signal that was in this 80 ms window generally continued throughout the unrelated speech, and was therefore potentially related to it. This resulted in the exclusion of 795 annotations. No annotations were made when there was insufficient facial signal data due to head movements preventing full visibility or due to occlusions. Similar to the questions and responses, interrater reliability between the coders was calculated with raw agreement (*agr*; [70,71]) and a modified Cohen’s kappa (*k;* [72]) using a standard overlap criterion of 60%. In addition, we computed convergent reliability for annotation timing by using a Pearson’s correlation (*r*), standard error of measurement (*SeM*), and the mean absolute difference (*Mabs*, in ms) of signal onsets, to access how precise these annotations were in terms of timing, if there was enough data to compare. One question and one response in one of the three parts were selected randomly for each participant in all dyads (roughly equivalent to 1% of the data). This enabled us to compare all coders in a pairwise fashion on the same data. We excluded eyebrow lowering and mouth corners pulled back from all further analyses, since the paired comparisons including unmatched annotations showed low raw agreement and kappa scores. For all other facial signals, the paired comparisons showed an average raw agreement of 76% (*min* = 70%, *max* = 82%) and an average kappa of 0.96 (*min =* 0.94, *max* = 0.97), indicating almost perfect agreement.

Reliability for each individual facial signal was calculated to obtain a more detailed view on how reliable coders were for each specific facial signal. Non-reliable measurements because of insufficient data were excluded when calculating these averages (e.g., if there was not enough data to perform correlations or standard error of measurements between two coders). Results are shown in Table 1.

There was almost perfect agreement (*k >* 0.81) for eyebrow frowns, raises, frown raises, unilateral raises, eye widenings, squints, blinks, gaze shifts, nose wrinkles, pressed lips, and smiles. There was a substantial agreement (*k >* 0.61) for mouth corners down. When there was enough data to perform a Pearson’s correlation, all signals showed *r* = 1 with a *p* < 0.0001, indicating a strong correlation. There was not enough data to perform a correlation for eyebrow frown raises, nose wrinkles, and mouth corners down.

The measurement unit for the standard error of measurement was in milliseconds, and one video frame was equivalent to 40 ms. Thus, we considered the variance based on the reliability of the signals (as showed by *SeM*) as very low when *SeM* < 40, low when *SeM* < 80, moderate *SeM* < 160, and high *SeM* < 160. There was a very low variance for the coding of eyebrow unilateral raises, squints, blinks, gaze shifts, nose wrinkles, pressed lips, and mouth corners down. A low variance was found for and eyebrow raises and eye widenings. A moderate variance was found for eyebrow frowns, frown raises, and a high variance was found for smiles. The same rationale as the standard error of measurement was applied for the mean absolute difference: a very precise annotation timing was found for blinks and nose wrinkles (*Mabs* < 40), a precise annotation timing for eyebrow unilateral raises and squints (*Mabs* < 80), a moderate annotation timing for eyebrow raises, frown raises, eye widenings, gaze shifts, mouth corners down (*Mabs* < 160). A poor annotation timing was found for eyebrow frowns, pressed lips, and smiles (*Mabs* > 160). The complete list of all reliability pairwise comparisons per coder and per signal, as well as the reliability script can be found on the Open Science Framework project website https://osf.io/x89qj/ (last accessed on 28 July 2021).

An overview of the final list of facial signals with durations per signal can be found in Table 2.

### 2.4. Analysis

The main results of our study are descriptive in nature; therefore, they do not contain statistical tests. We do provide a clustering analysis for which standard statistical methods are used.

#### 2.4.1. Distribution of Facial Signals across Questions and Responses

Our first analyses aimed to quantify and describe how facial signals distribute across questions and responses. To quantify the proportional distribution of facial signals across questions and responses, we first calculated how many facial signals of each type occurred together with questions out of the respective signal’s total number of occurrences, and we did the same for responses. With this analysis, we asked whether, when a signal occurred during a question or response, it is more likely to occur in one rather than the other.

Second, we calculated the proportional distribution of questions and responses across the different types of facial signals. To do so, we calculated how many out of all questions occurred together with a particular facial signal, and we did the same for responses. Here, the proportion of questions and responses contained any number of occurrences of a particular facial signal (e.g., multiple occurrences of a facial signal in a question or response were counted as one in that specific utterance). With this analysis, we asked how likely a given question or response was to contain *a* particular signal out of all questions or responses.

#### 2.4.2. Clustering of Facial Signals within Questions and Responses

For the clusters, we aimed to identify specific combinations of co-occurring facial signals that map onto questions and responses. We did this by looking at combinations using three different approaches. First, we looked at the frequency with which pairs of signals co-occur in questions and responses. Then, we tested whether there were any particular facial signals that are statistically predictive (or strongly associated) with an utterance being a question or a response, and are therefore able to reliably differentiate between questions and responses based on their occurrence frequency. Finally, we assessed whether a particular set of signals is characteristic of questions and responses.

For the first approach, we determined which pairs of facial signals frequently occurred together by analysing two data frames. One consisted of questions x facial signals, and the other of responses x facial signals. In these data frames, each row was either a question or response and each column the number of facial signals overlapping with that specific utterance. This provides a quantification of how frequently each pair of facial signals occurred together. This was performed as a test to see if there were any frequent co-occurrences of facial signals at all before examining potential clusters.

For the second approach, in order to find out if there are particular facial signals that differentiate questions from responses, we employed Decision Tree (DT) models [75]. DT models determine the groupings of (or single) facial signals that are strongly associated with (i.e., statistically predictive of) an utterance being either a question or a response. The purpose of this step was to determine whether there is any evidence that the two social actions are distinguishable based on the set of facial signals that accompany them. DT models consist of machine-learning methods to construct prediction models using continuous or categorical data. Based on the input data, DT models build logical “if... then” rules to predict the input cases. The models come from partitioning the data space in a recursive way, fitting a prediction model for each partition, which is represented in a DT. In this analysis, partitioning meant finding the specific configuration of facial signal combinations that predicted whether the utterance was a question or a response. We used conditional inference (CI; [76]) with holdout cross-validation, since CI selects on the basis of permutation significance tests which avoids the potential variable selection bias in similar decision trees and lead to the most optimal pruned decision tree. Cross-validation is a technique used to split the data into training and testing datasets, and holdout is the simplest kind as it performs the split only once [77]. To this end, we analysed a data frame consisting of utterances x facial signals. Each row was either a question or a response, and each column indicated occurrence of a specific facial signal with a 0 (not present) or 1 (present). One additional column indicated the utterance category (question or response). To test the statistical significance of the classification analysis, we used permutation tests [78], which are non-parametric methods for hypothesis testing without assuming a specific distribution [79]. This permutation shuffles the dataset to calculate accuracies a repeated number of times, and is compared to the actual accuracy without shuffling. The *p*-value was obtained from calculating the percentage of cases where the random shuffle gave higher accuracies than the actual accuracy. We used the same data and holdout cross-validation as in previous classification analysis, and repeated the simulation a 1000 times.

For the third approach, after determining whether there were particular facial signals that are statistically predictive with an utterance being a question or a response, we asked whether there were specific combinations of signals that occurred within questions and responses using Multiple Correspondence Analysis (MCA; [80]). MCA is the application of correspondence analysis (CA) to categorical variables and enables one to summarise relationships between variables, similar to Principle Component Analysis (PCA) but more suitable to represent non-continuous distances between variable categories in the factorial space. The (squared) distance between facial signals is calculated based on how much they have in common. In other words, signals that frequently co-occur in either questions or responses should cluster together with shorter distances, with distinct clusters when different sets of signals occur together. We analysed two data frames. One consisted of questions x facial signals, and the other of responses x facial signals. In both data frames, each row was either a question or a response, and each column indicated occurrence of a specific facial signal with a 0 (not present) or 1 (present). We first plotted the cloud of facial signal variables by projecting it on orthogonal axes to visualise their similarity or dissimilarity using their (squared) distance. Then, we summarised the similarities between facial signals in dendrograms, or trees of categorical variable groups, to show what the cluster partitions contained and at what point the facial signals were merged. The distance between clusters is represented in the dendrograms by the height between facial signals. The smaller the height at which two facial signals are joined together, the more similar they are. The bigger the height, the more dissimilar. To test the optimal number of cluster partitions, bootstrap samples of the trees of categorical variable groups (*n =* 22) were created to produce stability plots. Stability plots tell us at which number the MCA clustering solution is optimal. The dendrogram was cut to the optimal clustering number to see in which clusters each variable should be allocated [81].

#### 2.4.3. Timing of Facial Signals within Questions and Responses

In order to study whether facial signals occur primarily early or late, and whether there were differences in questions and responses, we first looked at the difference in proportion of facial signals with an onset before the start of a question or response and after the start of a question or response by splitting the data in two data frames. The first consisted of facial signals with an onset before the start of a question or response, the second consisted of facial signals after the start of a question or response. The split in pre-onset and post-onset data frames was only used in this first analysis. As a second analysis, we plotted the onset of facial signals relative to the onset of questions and responses, to see where the facial signals started relative to the utterance onset. Finally, to get a better idea of how the facial signal onsets distributed within the utterances, utterance duration was standardised between 0 (onset utterance) and 1 (offset utterance), and facial signal onsets were plotted relative to that number.

#### 2.4.4. Analysis and Session Information

The analyses were conducted in *R* (3.6.1; [82]) with *RStudio* (1.2.5019; [83]) using additional packages *PredPsych* (0.4; [77]), *FactoMineR* (2.3; [84]), and *ClustOfVar* (1.1; [81]). Moreover, we used *tidyr* (1.0; [85]), *plyr* (1.8.4; [86]), *dplyr* (1.0.2; [86]), *stringr* (1.4; [87]), *reshape2* (1.4.4; [88]), *purrr* (0.3.3; [89]), *forcats* (0.4.0; [90]), *caret* (6.0—86; [91]), and *car* (3.0—10; [92]). For visualization, we used packages *ggplot2* (3.2.1; [93]), *factoextra* [94]), *gridExtra* (2.3; [95]), *viridis* (0.5.1; [96]), and *scales* (1.0.0; [97]). The analysis script and additional session information can be found on the Open Science Framework project website https://osf.io/x89qj/ (last accessed on 28 July 2021).

## 3. Results

### 3.1. Distribution of Facial Signals across Questions and Responses

To quantify the distribution of facial signals across questions and responses, we first looked at the proportion of all occurrences of a facial signal that occur in questions and responses (i.e., also taking into account multiple occurrences of a signal within one question or response). With this analysis, we asked whether, when a signal occurs during a question or response, it is more likely to occur in one rather than the other. There were 13,214 facial signals that accompanied questions and 10,868 facial signals that accompanied responses. Specifically, we found that out of 1269 eyebrow frowns, 71% (*n =* 895) co-occurred with questions and 39% (*n =* 374) with responses. Out of 2832 eyebrow raises, 53% (*n =* 1503) co-occurred with questions and 47% (*n =* 1329) with responses. Out of 233 frown raises, 59% (*n =* 138) co-occurred with questions and 41% (*n =* 95) with responses. Out of 344 unilateral raises, 59% (*n =* 204) co-occurred with questions and 41% (*n =* 140) with responses. Out of 446 eye widenings, 64% (*n =* 286) co-occurred with questions and 36% (*n =* 160) with responses. Out of 1172 squints, 66% (*n =* 771) co-occurred with questions and 34% (*n =* 401) with responses. Out of 9582 blinks, 53% (*n =* 5033) co-occurred with questions and 47% (*n =* 4549) with responses. Out of the 5193 gaze shifts away from the interlocutor that accompanied questions and responses, 51% (*n =* 2642) co-occurred with questions and 49% (*n =* 2551) with responses. Out of 138 nose wrinkles, 63% (*n =* 87) co-occurred with questions and 37% (*n =* 51) with responses. Out of 101 pressed lips, 45% (*n =* 45) co-occurred with questions and 55% (*n =* 56) with responses. Out of 91 mouth corners down, 53% (*n =* 48) co-occurred with questions and 47% (*n =* 43) with responses. Lastly, out of 2681 smiles, 58% (*n =* 1562) co-occurred with questions and 42% (*n =* 1119) with responses (Figure 2).

Second, we looked at the proportion of questions and responses that contained (any number of occurrences of) a particular facial signal. With this analysis, we asked how likely a given question or response was to contain *a* particular signal out of all questions or responses. Out of all 6778 questions, 13% (*n =* 856) were accompanied with eyebrow frowns, 20% (*n =* 1343) with raises, 2% (*n =* 136) with frown raises, and 3% (*n =* 189) with unilateral raises. Moreover, 4% (*n =* 276) were accompanied with eye widenings, 11% (*n =* 740) with squints, 51% (*n* = 3454) with blinks, and 35% (*n =* 2402) with gaze shifts. Furthermore, 1% (*n =* 82) were accompanied with nose wrinkles, 1% (*n =* 45) with pressed lips, 1% (*n =* 46) with mouth corners down, and 23% (*n =* 1528) with smiles. Out of all 4553 responses, 8% (*n =* 358) were accompanied with eyebrow frowns, 25% (*n* = 1145) with raises, 2% (*n* = 91) with frown raises, and 3% (*n* = 128) with unilateral raises. Moreover, 3% (*n =* 151) were accompanied with eye widenings, 8% (*n =* 375) with squints, 59% (*n =* 2679) with blinks, 49% (*n =* 2244) were accompanied with gaze shifts. Furthermore, 1% (*n* = 50) were accompanied with nose wrinkles, 1% (*n* = 54) with pressed lips, 1% (*n* = 43) with mouth corners down, and 24% (*n* = 1089) with smiles (Figure 3).

### 3.2. Clustering of Facial Signals within Questions and Responses

#### 3.2.1. Co-Occurrences between Facial Signals

Facial signals do not always occur in isolation; therefore, we aimed to identify specific *combinations* of co-occurring facial signals that map onto questions and responses. We first determined if groups of two facial signals frequently occurred together, to see if there were any groupings of facial signals at all before examining potential clusters. For both questions and responses, there was a high number of eyebrow frowns with squints, eyebrow raises with eye widenings, and eyebrow raises with smiles. Moreover, there was a high number of blinks with eyebrow frowns, raises, squints, gaze shifts, and smiles. Furthermore, there was a high number of gaze shifts with eyebrow raises, and with smiles. There was a higher number of co-occurrences in questions for eyebrow frowns with blinks, and squints with blinks. However, there was a higher number of co-occurrences in responses for blinks with eyebrow raises, gaze shifts, and smiles. Additionally, there were also more gaze shifts with eyebrow raises and with smiles. Overall, the largest number of co-occurrences between facial signals was found in responses (Figure 4).

#### 3.2.2. Decision Tree Models

Before analysing any clusters of signals in questions and responses, we wanted to explore whether it was possible to distinguish between a question and a response based on groupings (or single) facial signals. We employed DT models, which constructed prediction models from specific configurations of facial signal combinations to statistically predict whether a verbal utterance was more likely to be a question or a response. With this analysis, we wanted to determine whether there is any evidence that the two social actions are distinguishable based on the frequency with which (a subset of) facial signals accompanied them. This analysis was performed on 11,331 observations. Results showed eight terminal nodes. A main pattern that can be gleaned from the tree is that eyebrow frowns appear to be amongst the most powerful visual question markers, since they are associated with the highest confidence values both when they occurred in the absence and in the presence of gaze shifts. Another pattern is that the verbal utterance was statistically predicted to be a question in all cases, except when there were gaze shifts with eyebrow raises. In that case, the verbal utterance was predicted to be a response. Although all other combinations of facial signals were predicted by the model to be questions, the confidence of this prediction changed depending on the combination. For instance, gaze shifts with blinks resulted in a 50% chance of being a question or a response (Figure 5). The permutation tests (number of simulations = 1000) showed an overall accuracy of 61% on the dataset, similar to accuracies obtained using the same type of model [98,99,100], with *p* = 0.001, suggesting a significant classification accuracy.

#### 3.2.3. Multiple Correspondence Analysis

After determining whether questions and responses could be distinguished based on facial signals, we asked whether there were specific combinations of signals that occur within questions and responses by looking at the likelihood that particular facial signals occur with one another, irrespective of their frequency. The MCA analysis consisted of four steps. First, relationships were summarised between facial signals by using their (squared) distance, which was calculated based on how frequently they co-occurred in either questions or responses. The MCA analysis was performed on 7934 observations (4746 for questions + 3188 for responses), using 70% of the data for training. Like PCA and CA, we represented the cloud of variables by projecting it on orthogonal axes (Figure 6).

Second, the similarities between facial signals were summarised in dendrograms, or trees of categorical variable groups, to show what the cluster partitions contained and at what point the facial signals were merged together as a cluster. Third, in order to determine how many distinct clusters occur, bootstrap samples of the trees (*n =* 22) were created to produce stability plots. These plots suggest that the 12 variables of the MCA clustering can be combined into eight optimal groups of variables for both questions and responses, as the curve stops increasing around eight clusters (Figure 7).

Lastly, the fourth step involved cutting the dendrograms to the optimal eight-cluster solution to see in which clusters each variable should be allocated. The ordering of the facial signals is not the same between questions and responses, but when looking below the green horizontal line indicating the eight-cluster solution (Figure 8), questions and responses show the same stable clusters. Both questions and responses consist of the following clusters: (1) blinks and gaze shifts; (2) eyebrow frowns, squints, and nose wrinkles; and (3) eyebrow raises and eye widenings. Eyebrow frown raises, eyebrow unilateral raises, pressed lips, mouth corners down, and smiles did not form reliable clusters.

To summarise, when looking at the frequency with which pairs of signals co-occur in questions and responses, there are several pairings between facial signals that occur frequently, some of which are more typical for questions and other for responses. Moreover, the DT models show that it is possible to statistically differentiate between questions and responses based on facial signals. Specifically, eyebrow frowns were predicted with most confidence to mark questions (even if co-occurring with gaze shifts), and gaze shifts with eyebrow raises were predicted by the model to be most likely in responses. Thus, questions and responses appear to be different in terms of individual facial signals. Finally, the MCA shows that, without taking into account relative frequency differences between questions and responses, the formal clusters between questions and responses are the same, and indicates that the signals that are likely to co-occur with one another are: (1) blinks and gaze shifts; (2) eyebrow frowns, squints, and nose wrinkles; and (3) eyebrow raises and eye widenings. These three clusters are stable combinations of signals within questions and responses, despite the frequency of these clusters occurring within each of these two social actions being different.

### 3.3. Timing of Facial Signals within Questions and Responses

To study the timing of facial signals, we first looked at the difference in proportion of facial signals with an onset before the start of a question or response and after the start of a question or response. We split facial signals with an onset before the start of a question or response and facial signals with an onset after the start of a question or response in two data frames, to better visualise their distribution before and after the start of an utterance. Seven out of twelve facial signals had a median onset equal to or after the start of the utterance (i.e., difference between onsets equal to or larger than 0 ms) for both questions and responses. Facial signals that mostly had an onset before the start of questions were eyebrow frowns (Q*n* = 473), frown raises (Q*n* = 74), gaze shifts (Q*n* = 1456), and smiles (Q*n* = 903). Facial signals that had an onset mostly after or at the start of questions were eye widenings (Q*n* = 148), squints (Q*n* = 421), and blinks (Q*n* = 3788). Other facial signals occurring after or at the start of questions were eyebrow unilateral raises (Q*n* = 152), nose wrinkles (Q*n* = 51), pressed lips (Q*n* = 41), and mouth corners down (Q*n* = 34). Facial signals that mostly had an onset before the start of responses were eyebrow raises (R*n* = 701), gaze shifts (R*n* = 1505), and smiles (R*n* = 709). Facial signals that had an onset mostly after or at the start of responses were eye widenings (R*n* = 99), squints (R*n* = 275), and blinks (R*n* = 3522). Other facial signals occurring after or at the start of responses were eyebrow frowns (R*n* = 236), frown raises (R*n* = 50), unilateral raises (R*n* = 99), nose wrinkles (R*n* = 30), pressed lips (R*n* = 46), and mouth corners down (R*n* = 32) (Table 3).

To see where the facial signals started with regard to the utterance onset (no grouping in a data frame before the start of the utterance and data frame after the start of the utterance), we looked at the onset of facial signals relative to the onset of questions and responses. Overall, facial signals had an earlier onset in questions compared to responses (Q*min =* −18,599, Q*max =* 12,960, R*min =* −12,418, R*max =* 17,694) (Figure 9). Facial signals with an earlier onset in questions compared to responses were eyebrow frowns (Q*min =* −12,495, Q*max =* 8695, R*min =* −8757, R*max =* 14,134), frown raises (Q*min =* −6293, Q*max =* 4815, R*min =* −5844, R*max =* 8988), eye widenings (Q*min =* −6600, Q*max =* 7356, R*min = −*2302, R*max =* 8139), squints (Q*min =* −8273, Q*max =* 8040, R*min =* −6996, R*max =* 10,792), blinks (Q*min =* −1080, Q*max =* 12,960, R*min =* −1142, R*max =* 17,694), and nose wrinkles (Q*min =* −1320, Q*max =* 2591, R*min =* −1320, R*max =* 9656). Facial signals with a later onset in questions were eyebrow raises (Q*min =* −18,599, Q*max =* 10,308, R*min =* −10,647, R*max =* 13,757), unilateral raises (Q*min =* −3146, Q*max =* 11,725, R*min =* −3485, R*max =* 13,894), gaze shifts (Q*min =* −7000, Q*max =* 10,760, R*min = −*12,418, R*max =* 16,894), pressed lips (Q*min =* −1029, Q*max =* 4440, R*min =* −2061, R*max =* 5200), mouth corners down (Q*min =* −1320, Q*max =* 3937, R*min =* −2304, R*max =* 6225), and smiles (Q*min =* −12,840, Q*max =* 7977, R*min = −*11,554, R*max =* 15,650) (Figure 10).

To see how the facial signal onsets distributed within the whole verbal utterances, utterance duration was standardised between 0 (onset utterance) and 1 (offset utterance), excluding pre- and post-utterance onsets. Most facial signals had an onset early in the utterance (QR*Mdn* < 0.50). Facial signals that had an onset early in the utterance were eyebrow frowns (Q*min =* 0, Q*max =* 1, R*min =* 0, R*max =* 0.99), eyebrow raises (Q*min =* 0, Q*max =* 1, R*min =* 0, R*max =* 0.99), frown raises (only Q*min* = 0.01, Q*max* = 0.97), unilateral raises (only R*min =* 0, R*max =* 1). Moreover, other early facial signals were eye widenings (Q*min =* 0, Q*max =* 0.95, R*min =* 0, R*max =* 0.98), squints (Q*min =* 0, Q*max =* 1, R*min =* 0, R*max =* 0.99), blinks (QR*min =* 0, QR*max =* 1) and gaze shifts (QR*min =* 0, QR*max =* 1), nose wrinkles (Q*min =* 0, Q*max* = 0.83, R*min =* 0, R*max* = 0.90), mouth corners down (only R*min =* 0.06, R*max* = 0.98). Facial signals that had an onset later in the utterance (QR*Mdn* > 0.50) were eyebrow frown raises (only R*min* = 0, R*max =* 0.98), unilateral raises (only Q*min =* 0, Q*max =* 1), pressed lips (Q*min =* 0.05, Q*max =* 1, R*min* = 0.04, R*max =* 1), mouth corners down (only Q*min =* 0.03, Q*max* = 0.99), and smiles (QR*min =* 0, QR*max =* 1).

Thus, similar to the onset of facial signals relative to the onset of questions and responses (Figure 9), the majority of facial signals had an earlier onset in questions compared to responses when looking at the timing of their onsets within the whole utterances (Figure 11). Facial signals with an earlier onset in questions compared to responses were eyebrow frown, raises, frown raises, eye widenings, squints, blinks, gaze shifts, and smiles. Facial signals with a later onset in questions compared to responses were eyebrow unilateral raises, pressed lips, and mouth corners down (Figure 12).

## 4. Discussion

In this study, we investigated a wide range of speech-accompanying conversational facial signals in a rich corpus of dyadic Dutch multimodal face-to-face interactions. We asked how the production of different facial signals mapped onto the communication of two fundamental social actions in conversation, questions and responses, by looking at their proportional distribution, clustering, and timing with regard to verbal utterance onset. Results showed a high proportion of facial signals being used, with, despite some overlap, a qualitatively different distribution in questions versus responses. Additionally, clusters of facial signals were identified within questions and responses. Importantly, most facial signals occurred early in the utterance, and had earlier onsets in questions than in responses. Below we discuss these findings in turn.

### 4.1. Distribution of Facial Signals across Questions and Responses

When looking at the first proportion of all question- or response-related facial signals, facial signals were more likely to co-occur with questions than with responses. It could be that questions are more visually marked because of a larger urgency for the listener to recognise the message fast to provide an appropriate answer, since long gaps indicate a dispreferred response [8]. Therefore, facial signals may facilitate the recognition of social actions such as questions early, which in turn may help potential following speakers to understand the intended message quickly and plan a timely response [9,10,11,12]. This result thus demonstrates that facial signals appear to form a core element of signalling speaker intentions in conversational social interaction.

Two particular interesting findings in regards to the distribution of facial signals in questions and responses are worth noting. The finding that there were more eyebrow movements in questions compared to responses is in line with past studies showing links between eyebrow movements and questions in spoken and signed languages [21,22,23,32,33,42,46,48,49,50,51,52,53,54,55,56]. This finding therefore supports the notion that eyebrow movements signal the intention to pose a question. A second notable finding is that, while most facial signals were more likely to occur in questions than responses, there was one exception: pressed lips had a higher proportion in responses compared to questions. This signal forms part of the not-face [37], suggesting that this facial expression may be more likely to be used to express negation or disagreement in responses.

Interestingly, when looking at the proportion of questions and responses that contained at least one of the facial signals analysed here, responses were more likely than questions to contain a facial signal, with the exception of eyebrow frowns and squints. This difference between the two proportion analyses for the distribution of facial signals in questions and responses indicates that there may be multiple facial signals per verbal utterance. In the first proportion analysis, we calculated how many facial signals of each type occurred together with questions out of the respective signal’s total number of occurrences, and we did the same for responses. This included multiple occurrences of facial signals. In the second proportion analysis, we calculated how many out of all questions occurred together with a particular facial signal, and we did the same for responses. In the second proportion analysis, multiple occurrences of a facial signal in a question or response were counted as *one* in that specific utterance. Thus, while responses may be more likely to have facial signals than questions, when questions do have a signal, they have more of it.

Overall, we found that speakers performed the highest proportion of total frequencies of facial signals in questions and responses for eyebrow frowns, raises, squints, blinks, gaze shifts, and smiles. This is in agreement with previous studies showing links between social actions and different facial signals such as eyebrow movements [21,22,23,24,25,31,32,33,37,38,42,46,48,49,50,51,52,53,54,55,56,57], squints [37], gaze shifts [25,39,47], and smiles [25,26,27,36].

In sum, these findings provide further evidence that these signals may be used to indicate different social actions such as questions and responses; thus, revealing the conversational intention of the speaker. This shows that different facial signals may critically contribute to the communication of social actions in naturalistic conversation, thus forming an integral part of human language.

### 4.2. Clustering of Facial Signals within Questions and Responses

When analysing clusters, we first observed several co-occurrences between facial signals within questions and responses. Facial signals that frequently co-occurred in both questions and responses were eyebrow frowns and squints, eyebrow raises and eye widenings, and eyebrow raises with smiles. Both blinks and gaze shifts (away from the interlocutor) frequently co-occurred with all other facial signals, but generally co-occurred more with other facial signals in responses. There was a higher number of co-occurrences in questions for eyebrow frowns with blinks, and squints with blinks. However, there was a higher number of co-occurrences in responses for blinks with eyebrow raises, gaze shifts, and smiles. Additionally, there were also more gaze shifts with eyebrow raises and with smiles. In general, the largest number of co-occurrences between facial signals were in responses. Blinks and gaze shifts were the most frequent facial signals overall, so it is not surprising that they co-occurred the most. These results show that there are specific pairwise groupings of co-occurring facial signals that distribute differently in questions versus responses (which may be accompanied by further signals).

To see whether it was possible to distinguish between a question or a response based on facial signals that accompanied them, we used DT models to construct prediction models. Results from the DT models showed that it is possible to statistically predict whether the utterance was a question or a response based on facial signals. Specifically, eyebrow frowns marked questions the most confidently (even if co-occurring with gaze shifts), and gaze shifts with eyebrow raises were predicted to often mark responses. This is in line with studies showing that eyebrow frowns are associated with questions in spoken and signed languages [21,22,23,31,32,33,46,50,51,53,54,55,57] and gaze shifts are associated with dispreferred responses [47]. It could be that eyebrow frowns often signal social actions that are subclasses of questions, to help potential following speakers to understand the intended message quickly in order to give a timely response [9,10,11,12]. The association between gaze shifts and responses may help the speaker indicate that their response to a question will not align with the social action that the form of the question projects [47], which may facilitate intention interpretation. Alternatively, it could be that there were more thinking-faces in responses [25,39], which could be used by the speaker to convey that they want to keep the floor until they have remembered what they were searching for, or may announce that they do not know something. However, it is also possible that at least a proportion of the gaze shifts were used as a turn-taking signal [102] or were associated with the cognitive planning of the responses [103,104]. This analysis demonstrates that a classification of different social actions based on a pool of facial signals as predictors is statistically possible, and that especially eyebrow frowns confidently marked questionhood with and without other facial signals.

After determining that questions and responses could be distinguished based on the co-occurring facial signals, we investigated combinations of facial signals characteristic for questions and responses by using MCA. This analysis indicated that the 12 facial signals could be clustered into eight stable clusters for both questions and responses. Both questions and responses consisted of the following clusters of facial signals: (1) blinks and gaze shifts; (2) eyebrow frowns, squints, and nose wrinkles; (3) eyebrow raises and eye widenings. The remaining facial signals consisted of eyebrow frown raises, eyebrow unilateral raises, pressed lips, mouth corners down, and smiles, each of which did not cluster with any other facial signals. The lack of clustering in the final five signals could be because they frequently associate with multiple facial signals from different clusters and therefore did not reliably fit with any one grouping. The second cluster (i.e., eyebrow frowns, squints, and nose wrinkles) has some resemblance to the facial expression that was previously identified in the literature as the not-face [37], which typically consists of eyebrow frowns, compressed chin muscles, and pressed lips, but was also found with squints and nose wrinkles. It could be that the third cluster in questions and responses indicates a ‘surprise-face’ [105], which fits well with our previous observation that this combination of facial signals was more frequent in questions when observing co-occurrences of facial signals (Figure 4). Lastly, the first cluster could have originated from the need to close the eyes before moving them to a different position for more stability. These findings show that there are specific constellations of facial signals that occur within both questions and responses, despite questions and responses differing in the frequency with which they were characterised by the occurrence of particular individual signals or signal combinations.

### 4.3. Timing of Facial Signals within Questions and Responses

Turning now to timing, the majority of facial signals happened early (before or at the very beginning of the verbal utterance). This confirms our hypothesis that facial signals may occur very early in the verbal utterance, or even prior to it, because this is how they may exert the greatest influence on early social action attribution: early visual signals may facilitate quick social action recognition for the following speakers and, thus, quick response planning, which is crucial for tight temporal coordination of conversational turn-taking [12]. This especially applies to the highly normed timing when responding to questions, since a longer than average gap may indicate a dispreferred response [8]. From a processing perspective, signalling one’s intention to ask a question early may therefore be particularly beneficial. Indeed, when looking at the overall distribution of facial signals across social actions, most had an earlier onset in questions compared to responses.

However, some signals occurred relatively late in the utterance too. This was the case for mouth movements, eyebrow frown raises and unilateral brow raises. It may be that the mouth movements were used for sarcastic or ironic intention, such as in studies on spoken and signed languages [25,26,27,28]. This intention is typically shown at the end of the utterance for a humoristic effect. The difference in timing of facial signals may also have occurred because of other factors. It could be that some facial signals indicated turn boundaries [22] or the begin or end of a topic [22,24], by either appearing at the start or the end of the speech. Facial signals at the start of the speech could indicate that the speaker intends to take the floor from a previous speaker, or gives the floor to a next speaker if signals are at the end of the utterance. Moreover, facial signals could have occurred at a specific point in the utterance prior to or following cognitive load [104,106]. However, with the corpus data we have analysed here where many layers of behaviour are inherently intertwined, we cannot tease apart the contributions of these other factors. Future experimental studies are required to tease these possibilities apart. Nevertheless, what we can conclude is that the distribution and timing of facial signal depend highly on the specific social action that is performed, and the patterns the present analysis has revealed are very much in line with a mechanism of early visual signalling which benefits fast social action ascription in conversation.

### 4.4. Limitations

The current study has some limitations. First, (extreme) laughter heavily affects the visibility of facial signals, since it involves many sub-movements, such as tilting back the head. Therefore, we coded the facial signals from the last evidence or until the first evidence of laughter. Although laughter occurred scarcely, it could be that the artificial cut-offs of facial signals in our data due to the occurrence of laughter still led to small artefacts in our calculation of timing. Second, multiple blinks sometimes occurred with gaps less than or equal to a single video frame (40 ms), which resulted in frames that showed only the white of the eyes, and made it difficult to make accurate annotations about blink onset or offset. Therefore, multiple blinks with gaps less than or equal to one frame were annotated as one blink. This could mean that more blinks were actually performed than we are reporting in this study. However, it could also be that multiple blinks following each other within one frame are perceived as a single communicative unit. Lastly, the present study focused on facial signals during questions and responses in dyadic conversations between acquaintances, but did not look at other categories of conversational social actions, sequential contexts other than question-response sequences, nor different intragroup and intergroup contexts.

### 4.5. Future Studies

Future studies investigating participants’ facial signalling in other dyadic contexts and across different cultures would show whether our findings are representative of questions and response more generally, or if they only apply to dyadic conversations between acquaintances in a particular setting. Additionally, including other social actions or a more fine-grained categorization of the broader social actions of questions and responses investigated here may help to elucidate the extent to which facial signals encode specific social intentions in conversation. It is of course possible—and in fact likely—that the particular patterns identified here are characteristic for certain social actions that questions and responses perform, but not for all of them. Identifying commonalities and differences at these more fine-grained levels of social actions remains an important avenue for future research, as well as trying to disentangle social action from turn-taking signals by looking at different social actions within the same sequential context and vice versa. Moreover, investigating the temporal organization of facial signals within an utterance in relation to one another would help to determine whether there is a fixed order of facial signals that characterises different social actions, including cases where they appear to form social-action-specific clusters of visual signals. In addition, investigating how facial signals temporally synchronise with speech (including prosodic patterns) could show how closely they are aligned during different social actions. Another aspect requiring future research is consideration of the detailed interactional processes that underpin the communication of questions and responses, in particular with respect to the extent to which interlocutors may shape current speakers’ facial signalling during questions and responses. Finally, investigating whether the neural signatures differ when social actions are carried by a combination of facial signals and words rather than verbal utterances alone would provide us with important insights into the role of co-speech facial signals during social action recognition.

## 5. Conclusions

In conclusion, this study showed that questions and responses are characterised by distinct distributions of facial signals, and consist of stable clusters of facial signals. Moreover, most facial signals occurred early in the turn. These findings suggest that specific facial signals, or combinations of facial signals, may be used to indicate different social actions, thus providing visual cues to speakers’ social intentions in conversation. The early timing of facial signals could provide a potential facilitative effect of facial signals for social action attribution, which in turn, may help potential following speakers to recognise the speaker’s intended message in a timely way during conversation; thus, facilitating fast responding. In sum, the results from this study highlight the potentially important role of the body in the pragmatics of human communication, and provide a foundation for investigating the cognitive and neural basis of social action recognition in face-to-face human communication.

## Figures and Tables

**Figure 1 brainsci-11-01017-f001:**
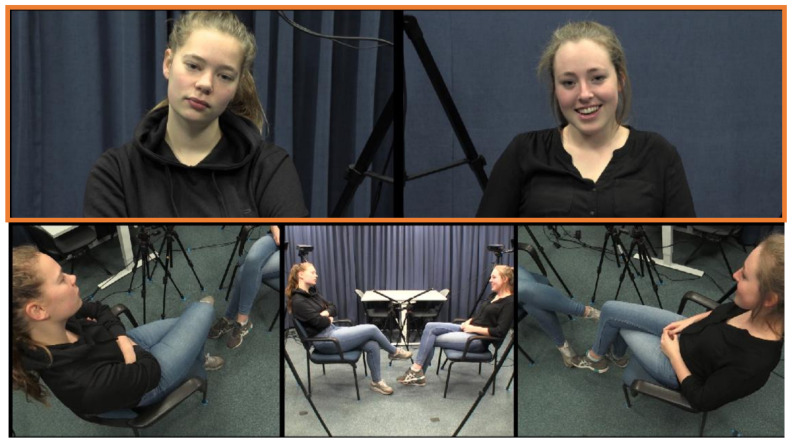
Still multiplex frame from one dyad. *Top panel:* frontal view, *bottom left and right panel:* bird view, *middle panel:* scene view. The orange frame indicates the camera angle used for the present analysis.

**Figure 2 brainsci-11-01017-f002:**
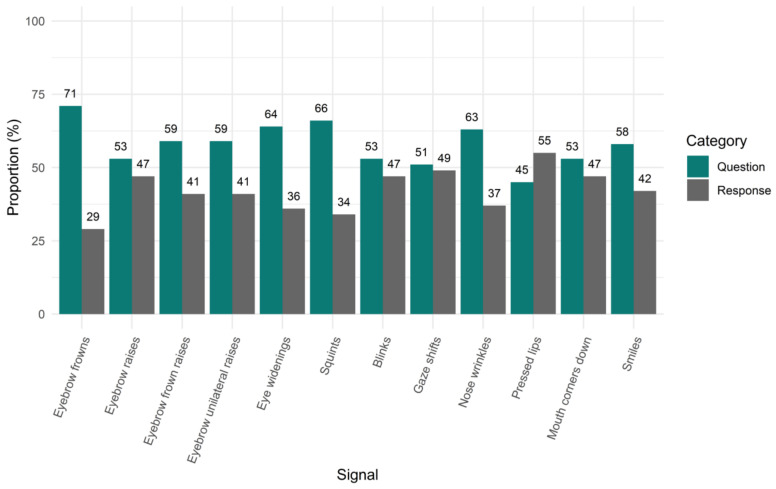
Proportion of facial signals in questions and responses. On the x-axis, we see facial signals split by question and response category. On the y-axis, the proportion is given for all occurrences of facial signals in questions and responses, taking into account multiple occurrences of a signal within one question or response.

**Figure 3 brainsci-11-01017-f003:**
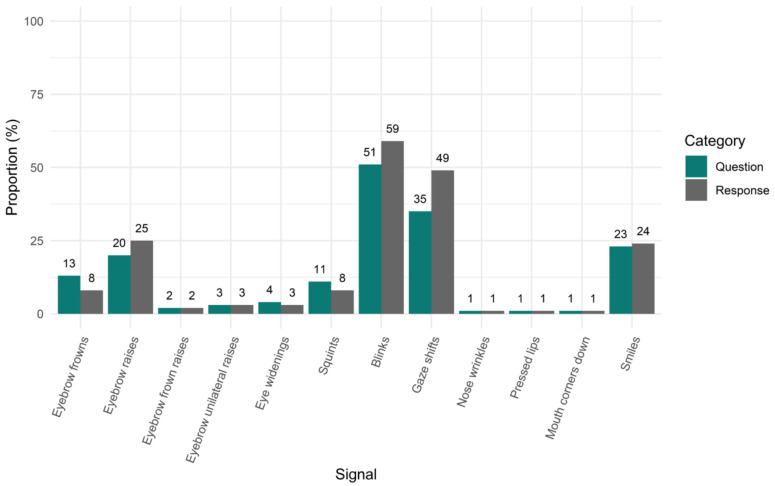
Proportion of questions and responses with facial signals. On the x-axis, we see facial signals split by question or response category. On the y-axis, the proportion is given of all questions or responses that contained (any number of occurrences of) a particular facial signal.

**Figure 4 brainsci-11-01017-f004:**
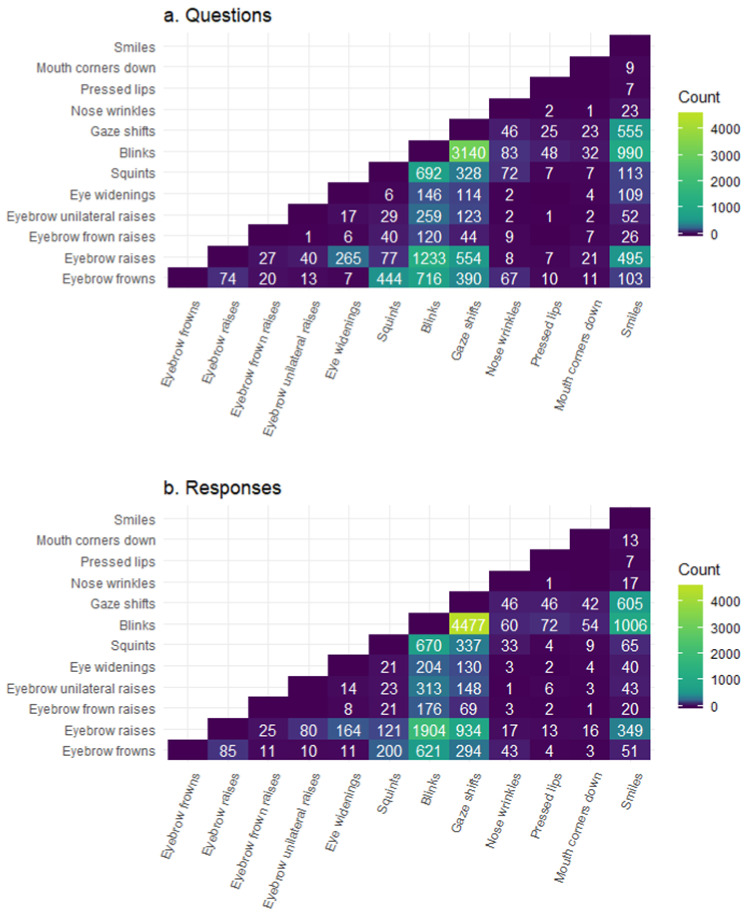
Co-occurrences of facial signals in questions (**a**) and responses (**b**). Count indicates the number of co-occurrences between two facial signals that overlap with a question (**panel a**) or a response (**panel b**). When two signals have no co-occurrences, the square is left blank.

**Figure 5 brainsci-11-01017-f005:**
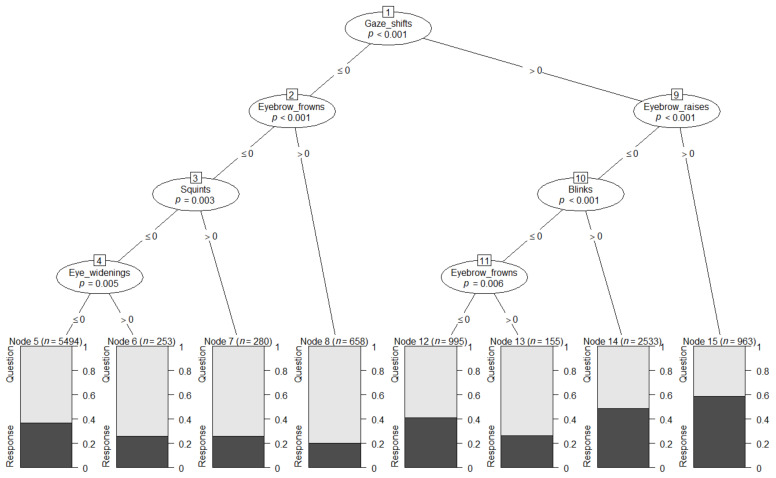
Conditional inference decision tree. The decision nodes are represented by circles, and each has a number. They show which facial signals are most strongly associated with the Bonferroni adjusted *p*-value of the dependence test. The input variable to split on is shown by each of these circles, which are divided sequentially (start at the top of the tree). The left and right branches show the cut-off value (i.e., ≤0 means no signals present, >0 signals present). The shaded area in the output nodes represents the proportion of response cases in that node, while the white area shows the proportion of question cases in that node. Therefore, output nodes that are primarily white indicate that an utterance would statistically be predicted to be a question, while a primarily shaded output node indicates a predicted response.

**Figure 6 brainsci-11-01017-f006:**
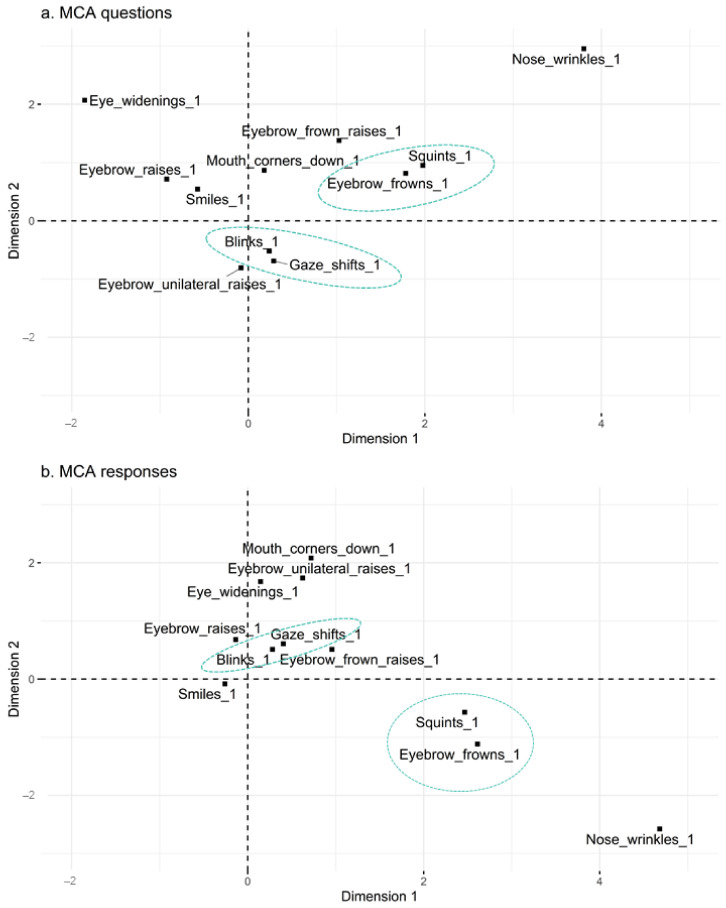
Plane representation of the cloud of variables for questions (**a**) and responses (**b**). Approximately 26% of the largest possible variance are provided with the first two principal components (dimension 1 + 2) for present facial signals indicated by suffix “_1” in questions and responses. The first principal component accounts for the largest possible variance in the dataset. The second principal component accounts for the next largest variance. The (squared) distance between facial signals gives a measure of their similarity or dissimilarity. Green dashed circles indicate which facial signals appear to be most closely related, and therefore co-occur the most.

**Figure 7 brainsci-11-01017-f007:**
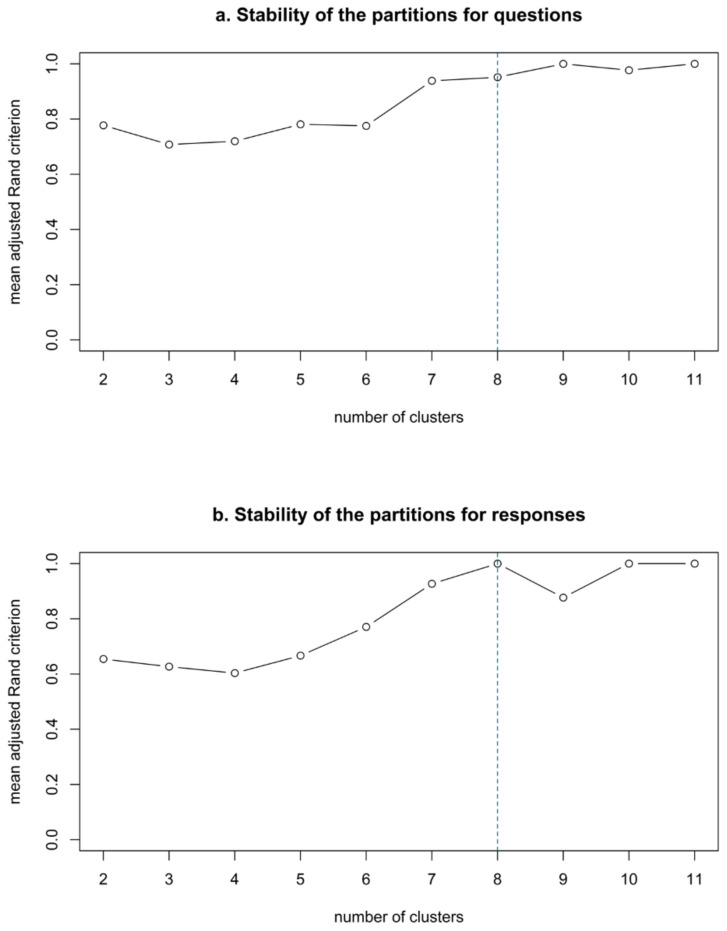
Stability plot for questions (**a**) and responses (**b**). This plot evaluates the stability of partitions from a hierarchy of variables, using bootstrap samples (*n* = 22) of the question and response trees. The mean of the corrected Rand indices measures the similarity between clusterings based on co-occurrences [101], and is plotted according to the number of clusters. The partitions are interpreted as stable when the curve stops increasing.

**Figure 8 brainsci-11-01017-f008:**
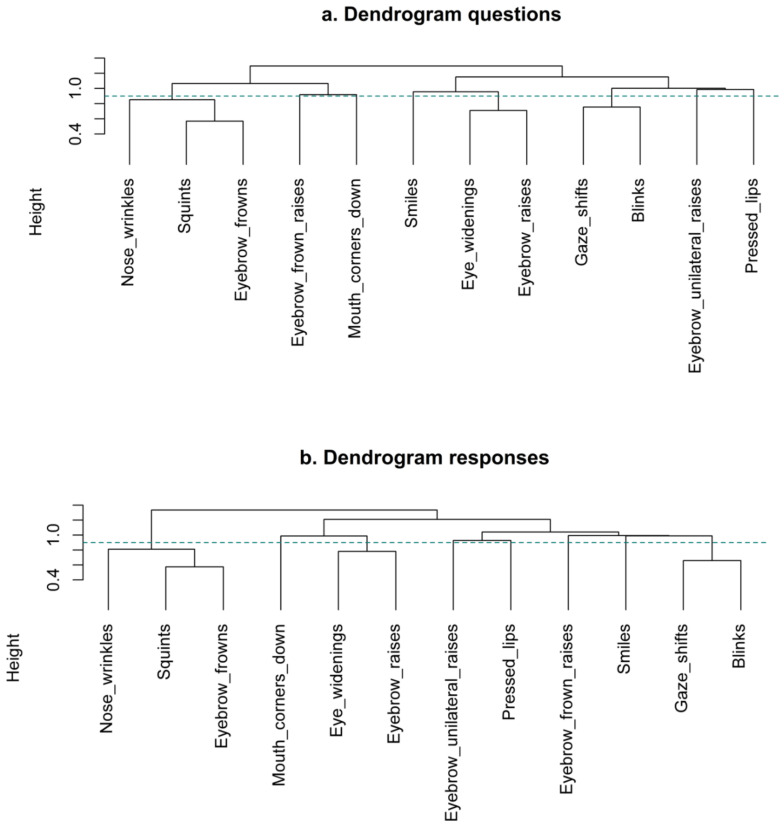
Cluster dendrogram of categorical variable groups for questions (**a**) and responses (**b**). In the dendrogram, the y-axis represents the distance between clusters. The smaller the height at which two facial signals are joined together, the more similar they are. The bigger the height, the more dissimilar. The horizontal bars indicate the point where the clusters are merged. The eight-cluster solution is indicated below the green horizontal dashed line through the dendrogram.

**Figure 9 brainsci-11-01017-f009:**
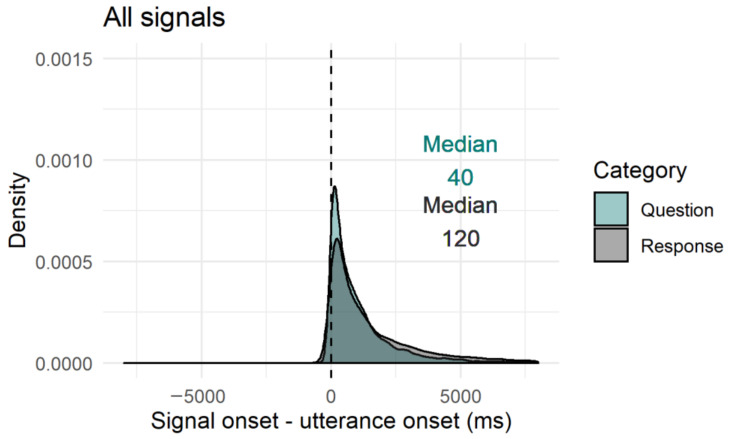
Overview of facial signals onset relative to verbal utterance onset. Question (green) and response (grey) median indicated in the figure. Negative values indicate that the signal onset preceded the start of the verbal utterance, ms = milliseconds.

**Figure 10 brainsci-11-01017-f010:**
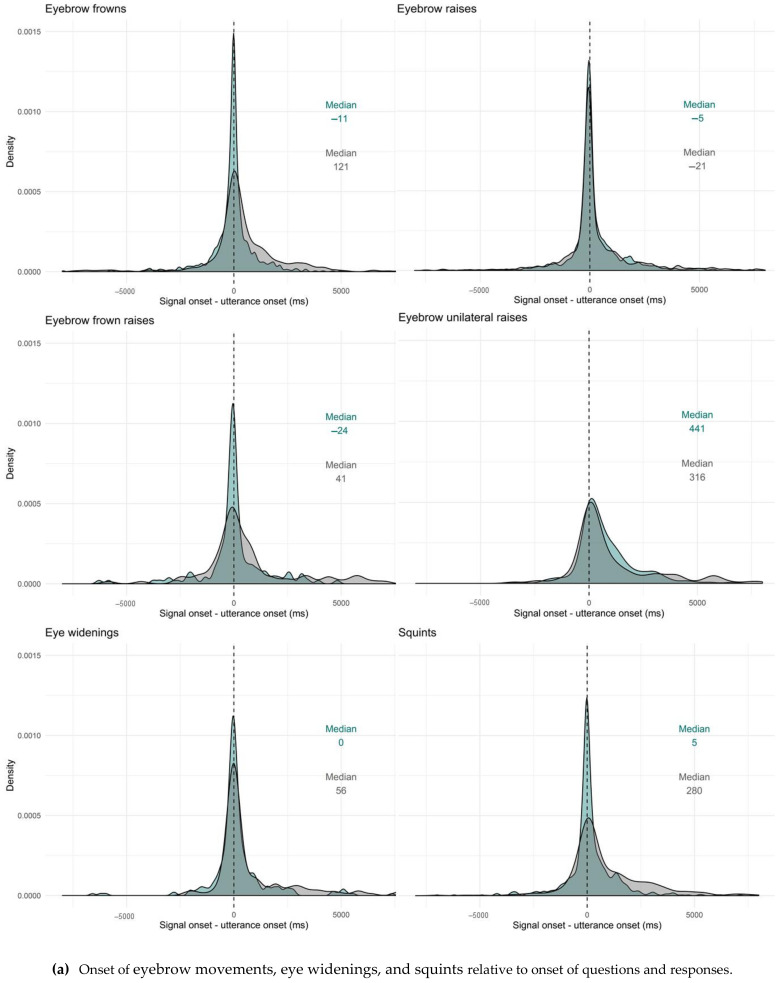
Onset of facial signals relative to onset of questions and responses (**panel a** and **b**). Question (green) and response (grey) median indicated in the figures. Negative values indicate that the signal onset preceded the start of the verbal utterance, ms = milliseconds.

**Figure 11 brainsci-11-01017-f011:**
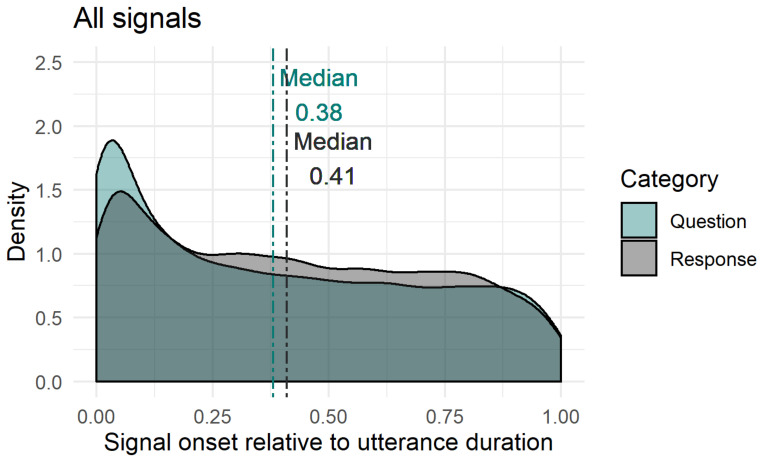
Overview of facial signals onset relative to standardised verbal utterance duration. Question (green) and response (grey) median indicated by dashed lines. This figure represents the facial signals onsets relative to the verbal utterance duration, therefore, pre- and post-utterance onsets were not included.

**Figure 12 brainsci-11-01017-f012:**
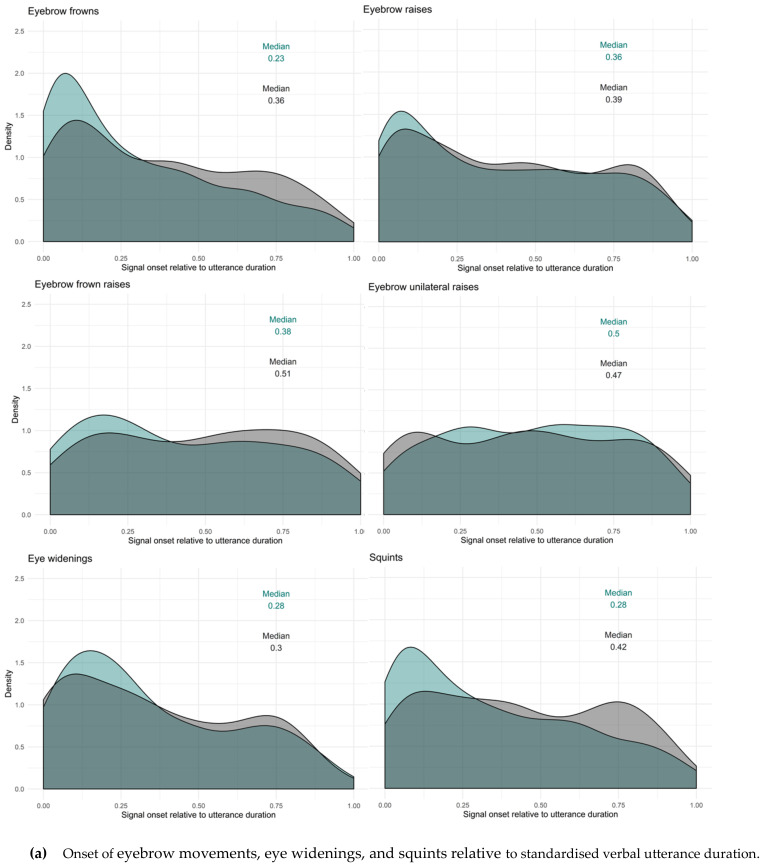
Facial signals onset relative to standardised verbal utterance duration (**panel a** and **b**). Question (green) and response (grey) median indicated in the figures. These figures represent the facial signals onsets relative to the verbal utterance duration, therefore, pre- and post-utterance onsets were not included.

**Table 1 brainsci-11-01017-t001:** Overview of facial signal reliability scores.

Signal	*agr*	*k*	*SeM*	*Mabs* (ms)
Eyebrow frowns	98%	0.90	84.97	167.58
Eyebrow raises	97%	0.97	46.07	120.44
Eyebrow frown raises	100%	0.83	97	132
Eyebrow unilateral raises	99%	0.88	13.49	46.57
Eye widenings	99%	0.83	46.30	129.16
Squints	99%	0.91	29.69	73
Blinks	92%	0.97	9.85	30.65
Gaze shifts	98%	0.99	36.89	112
Nose wrinkles	100%	0.81	24	40
Pressed lips	99%	0.86	34	380
Mouth corners down	97%	0.80	31	110
Smiles	97%	0.96	201.41	480.67

*Note. agr = raw agreement* [70,71], *k =* Cohen’s kappa [72], *SeM =* standard error of measurement, *Mabs =* mean absolute difference (ms).

**Table 2 brainsci-11-01017-t002:** Overview of facial signals and their duration.

Signal	Total Number	*Mdn* Duration (ms)	*min* Duration (ms)	*max* Duration (ms)	*IQR* Duration (ms)
Eyebrow frowns	1337	960	40	17,640	1320
Eyebrow raises	3138	640	40	20,120	990
Eyebrow frown raises	253	1080	120	9800	1520
Eyebrow unilateral raises	436	400	40	4760	410
Eye widenings	530	680	80	13,720	760
Squints	1294	920	80	10,240	1120
Blinks	16,734	280	40	2000	120
Gaze shifts	6749	920	40	16,120	1240
Nose wrinkles	164	520	120	3760	580
Pressed lips	380	620	120	4600	560
Mouth corners down	210	620	40	3480	600
Smiles	3188	1800	40	160,000	2040

*Note. Mdn* = median, *min* = minimum, *max* = maximum, *IQR =* interquartile range, ms = milliseconds.

**Table 3 brainsci-11-01017-t003:** Proportion of facial signals with an onset before or after the start of a question (Q) or a response (R).

Signal	Stats	Onset Signal < Onset Utterance		Onset Signal > Onset Utterance	
		Q	R	Q	R
Eyebrow frowns	*%*	53	37	47	63
	*Mdn*	−267	−329	230	601
	*min*	−12,495	−8757	0	0
	*max*	−1	−1	8695	14,134
Eyebrow raises	*%*	50	53	50	47
	*Mdn*	−200	−200	439	680
	*min*	−18,599	−10,647	0	0
	*max*	−1	−1	10,308	13,757
Eyebrow frown raises	*%*	54	47	46	53
	*Mdn*	−242	−346	340	788
	*min*	−6293	−5844	0	0
	*max*	−5	−14	4815	8988
Eyebrow unilateral raises	*%*	25	29	75	71
	*Mdn*	−156	−159	800	924
	*min*	−3146	−3485	0	0
	*max*	−1	−22	11,725	13,894
Eye widenings	*%*	48	38	52	62
	*Mdn*	−230	−180	319	434
	*min*	−6600	−2302	0	0
	*max*	−2	−1	7356	8139
Squints	*%*	45	31	55	69
	*Mdn*	−214	−324	362	968
	*min*	−8273	−6996	0	0
	*max*	−1	−2	8040	10,792
Blinks	*%*	25	23	75	77
	*Mdn*	−108	−111	688	960
	*min*	−1080	−1142	0	0
	*max*	−1	−1	12,960	17,694
Gaze shifts	*%*	55	59	45	41
	*Mdn*	−456	−528	344	563
	*min*	−7000	−12,418	0	0
	*max*	−1	−2	10,760	16,894
Nose wrinkles	*%*	41	41	59	59
	*Mdn*	−94	−185	198	403
	*min*	−1320	−1320	0	0
	*max*	−3	−2	2591	9656
Pressed lips	*%*	9	18	91	82
	*Mdn*	−360	−662	1048	496
	*min*	−1029	−2061	201	50
	*max*	−160	−266	4440	5200
Mouth corners down	*%*	29	26	71	74
	*Mdn*	−238	−333	1108	562
	*min*	−1320	−2304	0	0
	*max*	−5	−80	3937	6225
Smiles	*%*	58	63	42	37
	*Mdn*	−897	−881	588	672
	*min*	−12,840	−11,554	0	0
	*max*	−1	−1	7977	15,650

*Note: %* indicates the proportion of the signal (split between signals occurring with questions and those with responses) with an onset before the utterance onset (left two columns), or after the utterance onset (right two columns), *Mdn =* median, *min* = minimum, *max* = maximum (all in milliseconds).

## Data Availability

The complete list of all reliability pairwise comparisons per coder and per signal, the reliability script, the analysis script, and additional session information are openly available on the Open Science Framework project website https://osf.io/x89qj/. (accessed on 29 July 2021).

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
