# Peer review of "Facial Signals and Social Actions in Multimodal Face-to-Face Interaction"

_brainsci, 2021, doi:10.3390/brainsci11081017_

Round 1

Reviewer 1 Report

This manuscript presents a competently conducted study involving coding of facial movements during questions and answers in a corpus of video-recorded casual conversations.  As the authors acknowledge, this is exploratory research, which means that the conclusions can only be tentative.  In my view, the research question that is addressed is framed at too general level for the findings to provide more than a first step towards understanding of the complex pattern of data analysed here.

The specific focus of research is on facial signals associated with asking questions and giving answers.  The intention seems to be to identify diagnostic signals and patterns of signals associated with these two conversational actions. 

However, this raises two issues:

  1. Are the questions and answers sampled in this corpus representative of questions and answers more generally?  The participants were acquaintances discussing low stakes issues and the level of disagreement about them was not directly assessed.  Do the results generalise to all kinds of questions and answers in different dyadic, intragroup and intergroup contexts, where the topic and stake varies and where disagreement may carry other consequences?
  2. Are questions and answers really unitary categories of conversational action in any case?  It might have been a better starting point to focus on specific kinds of question-answer sequence rather than working from the presumption that questions and answers hang together in their specific characteristics.  Indeed, the predictive value of the identified combined cues reported in this study remains too low to support the claim that there is a clear and consistent distinction between questions and answers when studied at this level.  On the basis of the findings, a researcher who wanted to identify questions and answers based on the identified cues would have an unacceptably low hit rate of 61% (l. 568), even if the specific Q-A patterns in this selective data-set generalised perfectly.

The authors made the decision to investigate facial signals of individual parties to the conversational exchange separately (ll. 293-295), which means that questions about attunement to the other party's response are unaddressed.  Interpretation of the results would be facilitated by analysis not only of the addressee's immediate facial response to the signals, but also of the linguistic context for the delivery and receipt of the signal.  Again, a finer-grained analysis would be likely to yield richer interpretation of these data (as acknowledged by the authors, ll. 915-917).

The analysis of differences between questions and answers also seems to ignore the base rates of the various signals (ll. 367-371).  From the data reported here we cannot be sure whether questions increase or decrease the frequency of signals from base rate, whether answers are having the corresponding effect, or both.

As the authors acknowledge, there are also issues with interpreting the data about timing of signals.  There is a clear confound between the pre and post-onset analysis of questions and answers and giving and taking the floor (ll. 886-892).  Again, using comparison data where questions and answers are not being presented would help to disambiguate things.

In general, I felt that this was a decent start to a potentially interesting research program that allowed the authors to try out some statistical techniques for identifying patterns of facial signals and their timing in questions and answers.  The results unsurprisingly show that there are some differences between questions and answers but these differences are inconsistent across different instances of those categories.  What is now needed is further more intensive research to identify what specific patterns of signal are actually achieving within subsets of questions and answers in different dyadic contexts with different speaker-addressee relations.

Reviewer 2 Report

Review of manuscript: Facial signals and social actions in multimodal face-to-face interaction

In this manuscript from Naomi Nota and colleagues, the authors annotated videos from 34 face-to-face conversation episodes, focusing on how questions and responses (n = ~10k) are characterized by different types of facial expressions. The authors then analyzed the distributions, clustering (co-occurrence), and timing of facial expressions, and provided highly informative new insights, for the first time of its kind, into how facial expressions would occur during questions and responses.

Overall, I think this study is flawless given its design and analysis. I especially highly appreciate the tremendous effort from the authors to conduct the study, which would clearly require tons of hours of work. The manuscript is also very nicely written, especially regarding the rich & precise description of the methods taken. But I also have a few very minor suggestions for the authors to consider:

  • The scope of Brain Sciences is to publish studies about the ‘brain’. Thus, I am wondering if the authors could consider elaborating a bit in the introduction and discussion, concerning how this study (and this line of research) may contribute to a better outlook of the neurobiology of social communication (It definitely does). Note, however, that this special issue welcomes studies from psychology, thus it is not mandatory to follow this comment.
  • Could you provide a figure illustrating how the facial signals look like (with a representative frame or several frames for each signal). This might help readers better grasp the fine-grained difference between these facial signals.
  • For some reason all figure captions are now shown as 1) a title plus 2) a caption starting with 'Note'. I feel this is fairly unconventional and difficult to read.

Round 2

Reviewer 1 Report

I appreciate the authors' amendments but think that further data would be required to make this into a standalone contribution to the literature.